# Neural Adaptive Sequential Monte Carlo

**Shixiang Gu**[†‡]    **Zoubin Ghahramani**[†]    **Richard E. Turner**[†]
[†] University of Cambridge, Department of Engineering, Cambridge UK
[‡] MPI for Intelligent Systems, Tübingen, Germany
sg717@cam.ac.uk, zoubin@eng.cam.ac.uk, ret26@cam.ac.uk

## Abstract

Sequential Monte Carlo (SMC), or particle filtering, is a popular class of methods for sampling from an intractable target distribution using a sequence of simpler intermediate distributions. Like other importance sampling-based methods, performance is critically dependent on the proposal distribution: a bad proposal can lead to arbitrarily inaccurate estimates of the target distribution. This paper presents a new method for automatically adapting the proposal using an approximation of the Kullback-Leibler divergence between the true posterior and the proposal distribution. The method is very flexible, applicable to any parameterized proposal distribution and it supports online and batch variants. We use the new framework to adapt powerful proposal distributions with rich parameterizations based upon neural networks leading to Neural Adaptive Sequential Monte Carlo (NASMC). Experiments indicate that NASMC significantly improves inference in a non-linear state space model outperforming adaptive proposal methods including the Extended Kalman and Unscented Particle Filters. Experiments also indicate that improved inference translates into improved parameter learning when NASMC is used as a subroutine of Particle Marginal Metropolis Hastings. Finally we show that NASMC is able to train a latent variable recurrent neural network (LV-RNN) achieving results that compete with the state-of-the-art for polymorphic music modelling. NASMC can be seen as bridging the gap between adaptive SMC methods and the recent work in scalable, black-box variational inference.

## 1 Introduction

Sequential Monte Carlo (SMC) is a class of algorithms that draw samples from a target distribution of interest by sampling from a series of simpler intermediate distributions. More specifically, the sequence constructs a proposal for importance sampling (IS) [1, 2]. SMC is particularly well-suited for performing inference in non-linear dynamical models with hidden variables, since filtering naturally decomposes into a sequence, and in many such cases it is the state-of-the-art inference method [2, 3]. Generally speaking, inference methods can be used as modules in parameter learning systems. SMC has been used in such a way for both approximate maximum-likelihood parameter learning [4] and in Bayesian approaches such as the recently developed Particle MCMC methods [3].

Critically, in common with any importance sampling method, the performance of SMC is strongly dependent on the choice of the proposal distribution. If the proposal is not well-matched to the target distribution, then the method can produce samples that have low effective sample size and this leads to Monte Carlo estimates that have pathologically high variance [1]. The SMC community has developed approaches to mitigate these limitations such as resampling to improve particle diversity when the effective sample size is low [1] and applying MCMC transition kernels to improve particle diversity [5, 2, 3]. A complementary line of research leverages distributional approximate inference methods, such as the extended Kalman Filter and Unscented Kalman Filter, to construct better proposals, leading to the Extended Kalman Particle Filter (EKPF) and Unscented Particle Fil-

ter (UPF) [5]. In general, however, the construction of good proposal distributions is still an open question that severely limits the applicability of SMC methods.

This paper proposes a new gradient-based black-box adaptive SMC method that automatically tunes flexible proposal distributions. The quality of a proposal distribution can be assessed using the (intractable) Kullback-Leibler (KL) divergence between the target distribution and the parametrized proposal distribution. We approximate the derivatives of this objective using samples derived from SMC. The framework is very general and tractably handles complex parametric proposal distributions. For example, here we use neural networks to carry out the parameterization thereby leveraging the large literature and efficient computational tools developed by this community. We demonstrate that the method can efficiently learn good proposal distributions that significantly outperform existing adaptive proposal methods including the EKPF and UPF on standard benchmark models used in the particle filter community. We show that improved performance of the SMC algorithm translates into improved mixing of the Particle Marginal Metropolis-Hasting (PMMH) [3]. Finally, we show that the method allows higher-dimensional and more complicated models to be accurately handled using SMC, such as those parametrized using neural networks (NN), that are challenging for traditional particle filtering methods .

The focus of this work is on improving SMC, but many of the ideas are inspired by the burgeoning literature on approximate inference for unsupervised neural network models. These connections are explored in section 6.

## 2    Sequential Monte Carlo

We begin by briefly reviewing two fundamental SMC algorithms, sequential importance sampling (SIS) and sequential importance resampling (SIR). Consider a probabilistic model comprising (possibly multi-dimensional) hidden and observed states $\boldsymbol{z}_{1:T}$ and $\boldsymbol{x}_{1:T}$ respectively, whose joint distribution factorizes as $p(\boldsymbol{z}_{1:T}, \boldsymbol{x}_{1:T}) = p(\boldsymbol{z}_1)p(\boldsymbol{x}_1|\boldsymbol{z}_1) \prod_{t=2}^{T} p(\boldsymbol{z}_t|\boldsymbol{z}_{1:t-1})p(\boldsymbol{x}_t|\boldsymbol{z}_{1:t}, \boldsymbol{x}_{1:t-1})$. This general form subsumes common state-space models, such as Hidden Markov Models (HMMs), as well as non-Markovian models for the hidden state, such as Gaussian processes.

The goal of the sequential importance sampler is to approximate the posterior distribution over the hidden state sequence, $p(\boldsymbol{z}_{1:T}|\boldsymbol{x}_{1:T}) \approx \sum_{n=1}^{N} \tilde{w}_t^{(n)} \delta(\boldsymbol{z}_{1:T} - \boldsymbol{z}_{1:T}^{(n)})$, through a weighted set of $N$ sampled trajectories drawn from a simpler proposal distribution $\{\boldsymbol{z}_{1:T}^{(n)}\}_{n=1:N} \sim q(\boldsymbol{z}_{1:T}|\boldsymbol{x}_{1:T})$. Any form of proposal distribution can be used in principle, but a particularly convenient one takes the same factorisation as the true posterior $q(\boldsymbol{z}_{1:T}|\boldsymbol{x}_{1:T}) = q(\boldsymbol{z}_1|\boldsymbol{x}_1) \prod_{t=2}^{T} q(\boldsymbol{z}_t|\boldsymbol{z}_{1:t-1}, \boldsymbol{x}_{1:t})$, with filtering dependence on $\boldsymbol{x}$. A short derivation (see supplementary material) then shows that the normalized importance weights are defined by a recursion:

$$w(\boldsymbol{z}_{1:T}^{(n)}) = \frac{p(\boldsymbol{z}_{1:T}^{(n)}, \boldsymbol{x}_{1:T})}{q(\boldsymbol{z}_{1:T}^{(n)}|\boldsymbol{x}_{1:T})}, \ \tilde{w}(\boldsymbol{z}_{1:T}^{(n)}) = \frac{w(\boldsymbol{z}_{1:T}^{(n)})}{\sum_n w(\boldsymbol{z}_{1:T}^{(n)})} \propto \tilde{w}(\boldsymbol{z}_{1:T-1}^{(n)}) \frac{p(\boldsymbol{z}_T^{(n)}|\boldsymbol{z}_{1:T-1}^{(n)})p(\boldsymbol{x}_T|\boldsymbol{z}_{1:T}^{(n)}, \boldsymbol{x}_{1:T-1})}{q(\boldsymbol{z}_T^{(n)}|\boldsymbol{z}_{1:T-1}^{(n)}, \boldsymbol{x}_{1:T})}$$

SIS is elegant as the samples and weights can be computed in sequential fashion using a single forward pass. However, naïve implementation suffers from a severe pathology: the distribution of importance weights often become highly skewed as $t$ increases, with many samples attaining very low weight. To alleviate the problem, the Sequential Importance Resampling (SIR) algorithm [1] adds an additional step that resamples $\boldsymbol{z}_t^{(n)}$ at time $t$ from a multinomial distribution given by $\tilde{w}(\boldsymbol{z}_{1:t}^{(n)})$ and gives the new particles equal weight.[1] This replaces degenerated particles that have low weight with samples that have more substantial importance weights without violating the validity of the method. SIR requires knowledge of the full trajectory of previous samples at each stage to draw the samples and compute the importance weights. For this reason, when carrying out resampling, each new particle needs to update its ancestry information. Letting $a_{\tau,t}^{(n)}$ represent the ancestral index of particle $n$ at time $t$ for state $\boldsymbol{z}_\tau$, where $1 \leq \tau \leq t$, and collecting these into the set $A_t^{(n)} = \{a_{1,t}^{(n)}, ..., a_{t,t}^{(n)}\}$, where $a_{\tau-1,t}^{(i)} = a_{\tau-1,\tau-1}^{(a_{\tau,t}^{(i)})}$, the resampled trajectory can be denoted $\boldsymbol{z}_{1:t}^{(n)} = \{\boldsymbol{z}_{1:t-1}^{A_{t-1}^{(n)}}, \boldsymbol{z}_t^{(n)}\}$ where $\boldsymbol{z}_{1:t}^{A_t^{(i)}} = \{\boldsymbol{z}_1^{a_{1,t}^{(i)}}, ..., \boldsymbol{z}_t^{a_{t,t}^{(i)}}\}$. Finally, to lighten notation, we use the shorthand

$w_t^{(n)} = w(z_{1:t}^{(n)})$ for the weights. Note that, when employing resampling, these do not depend on the previous weights $w_{t-1}^{(n)}$ since resampling has given the previous particles uniform weight. The implementation of SMC is given by Algorithm 1 in the supplementary material.

## 2.1 The Critical Role of Proposal Distributions in Sequential Monte Carlo

The choice of the proposal distribution in SMC is critical. Even when employing the resampling step, a poor proposal distribution will produce trajectories that, when traced backwards, quickly collapse onto a single ancestor. Clearly this represents a poor approximation to the true posterior $p(z_{1:T}|x_{1:T})$. These effects can be mitigated by increasing the number of particles and/or applying more complex additional MCMC moves [5, 2], but these strategies increase the computational cost.

The conclusion is that the proposal should be chosen with care. The optimal choice for an unconstrained proposal that has access to all of the observed data at all times is the intractable posterior distribution $q_\phi(z_{1:T}|x_{1:T}) = p_\theta(z_{1:T}|x_{1:T})$. Given the restrictions imposed by the factorization, this becomes $q(z_t|z_{1:t-1}, x_{1:t}) = p(z_t|z_{1:t-1}, x_{1:t})$, which is still typically intractable. The bootstrap filter instead uses the prior $q(z_t|z_{1:t-1}, x_{1:t}) = p(z_t|z_{1:t-1}, x_{1:t-1})$ which is often tractable, but fails to incorporate information from the current observation $x_t$. A halfway-house employs distributional approximate inference techniques to approximate $p(z_t|z_{1:t-1}, x_{1:t})$. Examples include the EKPF and UPF [5]. However, these methods suffer from three main problems. First, the extended and unscented Kalman Filter from which these methods are derived are known to be inaccurate and poorly behaved for many problems outside of the SMC setting [6]. Second, these approximations must be applied on a sample by sample basis, leading to significant additional computational overhead. Third, neither approximation is tuned using an SMC-relevant criterion. In the next section we introduce a new method for adapting the proposal that addresses these limitations.

## 3 Adapting Proposals by Descending the Inclusive KL Divergence

In this work the quality of the proposal distribution will be optimized using the inclusive KL-divergence between the true posterior distribution and the proposal, $\mathrm{KL}[p_\theta(z_{1:T}|x_{1:T})||q_\phi(z_{1:T}|x_{1:T})]$. (Parameters are made explicit since we will shortly be interested in both adapting the proposal $\phi$ and learning the model $\theta$.) This objective is chosen for four main reasons. First, this is a direct measure of the quality of the proposal, unlike those typically used such as effective sample size. Second, if the true posterior lies in the class of distributions attainable by the proposal family then the objective has a global optimum at this point. Third, if the true posterior does not lie within this class, then this KL divergence tends to find proposal distributions that have higher entropy than the original which is advantageous for importance sampling (the exclusive KL is unsuitable for this reason [7]). Fourth, the derivative of the objective can be approximated efficiently using a sample based approximation that will now be described.

The gradient of the negative KL divergence with respect to the parameters of the proposal distribution takes a simple form,

$$-\frac{\partial}{\partial \phi}\mathrm{KL}[p_\theta(z_{1:T}|x_{1:T})||q_\phi(z_{1:T}|x_{1:T})] = \int p_\theta(z_{1:T}|x_{1:T})\frac{\partial}{\partial \phi}\log q_\phi(z_{1:T}|x_{1:T})dz_{1:T}.$$

The expectation over the posterior can be approximated using samples from SMC. One option would use the weighted sample trajectories at the final time-step of SMC, but although asymptotically unbiased such an estimator would have high variance due to the collapse of the trajectories. An alternative, that reduces variance at the cost of introducing some bias, uses the intermediate ancestral trees i.e. a filtering approximation (see the supplementary material for details),

$$-\frac{\partial}{\partial \phi}\mathrm{KL}[p_\theta(z_{1:T}|x_{1:T})||q_\phi(z_{1:T}|x_{1:T})] \approx \sum_t \sum_n \tilde{w}_t^{(n)}\frac{\partial}{\partial \phi}\log q_\phi(z_t^{(n)}|x_{1:t}, z_{1:t-1}^{A_{t-1}^{(n)}}). \quad (1)$$

The simplicity of the proposed approach brings with it several advantages and opportunities.

**Online and batch variants.** Since the derivatives distribute over time, it is trivial to apply this update in an online way e.g. updating the proposal distribution every time-step. Alternatively, when learning parameters in a batch setting, it might be more appropriate to update the proposal parameters after making a full forward pass of SMC. Conveniently, when performing approximate

maximum-likelihood learning the gradient update for the model parameters $\theta$ can be efficiently approximated using the same sample particles from SMC (see supplementary material and Algorithm 1). A similar derivation for maximum likelihood learning is also discussed in [4].

$$\frac{\partial}{\partial\theta}\log[p_\theta(\boldsymbol{x}_{1:T})] \approx \sum_t \sum_n \tilde{w}_t^{(n)} \frac{\partial}{\partial\theta}\log p_\theta(\boldsymbol{x}_t, \boldsymbol{z}_t^{(n)}|\boldsymbol{x}_{1:t-1}, \boldsymbol{z}_{1:t-1}^{A_{t-1}^{(n)}}). \tag{2}$$

---

**Algorithm 1** Stochastic Gradient Adaptive SMC (batch inference and learning variants)

---

**Require:** proposal: $q_\phi$, model: $p_\theta$, observations: $X = \{\boldsymbol{x}_{1:T_j}\}_{j=1:M}$, number of particles: $N$
  **repeat**
    $\{\boldsymbol{x}_{1:T_j}^{(j)}\}_{j=1:m} \leftarrow \text{NextMiniBatch}(X)$
    $\{\boldsymbol{z}_{1:t}^{(i,j)}, \tilde{w}_t^{(i,j)}\}_{i=1:N,j=1:m,t=1:T_j} \leftarrow \text{SMC}(\theta, \phi, N, \{\boldsymbol{x}_{1:T_j}^{(j)}\}_{j=1:m})$
    $\triangle\phi = \sum_j \sum_{t=1}^{T_j} \sum_i \tilde{w}_t^{(i,j)} \frac{\partial}{\partial\phi}\log q_\phi(\boldsymbol{z}_t^{(i,j)}|\boldsymbol{x}_{1:t}^{(j)}, \boldsymbol{z}_{1:t-1}^{A_{t-1}^{(i,j)}})$
    $\triangle\theta = \sum_j \sum_{t=1}^{T_j} \sum_i \tilde{w}_t^{(i,j)} \frac{\partial}{\partial\theta}\log p_\theta(\boldsymbol{x}_t^{(j)}, \boldsymbol{z}_t^{(i,j)}|\boldsymbol{x}_{1:t-1}^{(j)}, \boldsymbol{z}_{1:t-1}^{A_{t-1}^{(i,j)}})$   (optional)
    $\phi \leftarrow \text{Optimize}(\phi, \triangle\phi)$
    $\theta \leftarrow \text{Optimize}(\theta, \triangle\theta)$   (optional)
  **until** convergence

---

**Efficiency of the adaptive proposal.** In contrast to the EPF and UPF, the new method employs an analytic function for propagation and does not require costly particle-specific distributional approximation as an inner-loop. Similarly, although the method bears similarity to the assumed-density filter (ADF) [8] which minimizes a (local) inclusive KL, the new method has the advantage of minimizing a global cost and does not require particle-specific moment matching.

**Training complex proposal models.** The adaptation method described above can be applied to any parametric proposal distribution. Special cases have been previously treated by [9]. We propose a related, but arguably more straightforward and general approach to proposal adaptation. In the next section, we describe a rich family of proposal distributions, that go beyond previous work, based upon neural networks. This approach enables adaptive SMC methods to make use of the rich literature and optimization tools available from supervised learning.

**Flexibility of training.** One option is to train the proposal distribution using samples from SMC derived from the observed data. However, this is not the only approach. For example, the proposal could be trained using data sampled from the generative model instead, which might mitigate over-fitting effects for small datasets. Similarly, the trained proposal does not need to be the one used to generate the samples in the first place. The bootstrap filter or more complex variants can be used.

## 4   Flexible and Trainable Proposal Distributions Using Neural Networks

The proposed adaption method can be applied to any parametric proposal distribution. Here we briefly describe how to utilize this flexibility to employ powerful neural network-based parameterizations that have recently shown excellent performance in supervised sequence learning tasks [10, 11]. Generally speaking, applications of these techniques to unsupervised sequence modeling settings is an active research area that is still in its infancy [12] and this work opens a new avenue in this wider research effort.

In a nutshell, the goal is to parameterize $q_\phi(\boldsymbol{z}_t|\boldsymbol{z}_{1:t-1}, \boldsymbol{x}_{1:t})$ – the proposal's stochastic mapping from all previous hidden states $\boldsymbol{z}_{1:t-1}$ and all observations (up to and including the current observation) $\boldsymbol{x}_{1:t}$, to the current hidden state, $\boldsymbol{z}_t$ – in a flexible, computationally efficient and trainable way. Here we use a class of functions called Long Short-Term Memory (LSTM) that define a deterministic mapping from an input sequence to an output sequence using parameter-efficient recurrent dynamics, and alleviate the common vanishing gradient problem in recurrent neural networks [13, 10, 11]. The distributions $q_\phi(\boldsymbol{z}_t|\boldsymbol{h}_t)$ can be a mixture of Gaussians (a mixture density network (MDN) [14]) in which the mixing proportions, means and covariances are parameterised through another neural network (see the supplementary for details on LSTM, MDN, and neural network architectures).

# 5 Experiments

The goal of the experiments is three fold. First, to evaluate the performance of the adaptive method for inference on standard benchmarks used by the SMC community with known ground truth. Second, to evaluate the performance when SMC is used as an inner loop of a learning algorithm. Again we use an example with known ground truth. Third, to apply SMC learning to complex models that would normally be challenging for SMC comparing to the state-of-the-art in approximate inference.

One way of assessing the success of the proposed method would be to evaluate $\text{KL}[p(\boldsymbol{z}_{1:T}|\boldsymbol{x}_{1:T})||q(\boldsymbol{z}_{1:T}|\boldsymbol{x}_{1:T})]$. However, this quantity is hard to accurately compute. Instead we use a number of other metrics. For the experiments where ground truth states $\boldsymbol{z}_{1:T}$ are known we can evaluate the root mean square error (RMSE) between the approximate posterior mean of the latent variables ($\bar{\boldsymbol{z}}_t$) and the true value $\text{RMSE}(\boldsymbol{z}_{1:T}, \bar{\boldsymbol{z}}_{1:T}) = (\frac{1}{T}\sum_t (\boldsymbol{z}_t - \bar{\boldsymbol{z}}_t)^2)^{1/2}$. More generally, the estimate of the log-marginal likelihood ($\text{LML} = \log p(\boldsymbol{x}_{1:T}) = \sum_t \log p(\boldsymbol{x}_t|\boldsymbol{x}_{1:t-1}) = \sum_t \log(\frac{1}{N}\sum_n w_t^{(n)})$) and its variance is also indicative of performance. Finally, we also employ a common metric called the effective sample size (ESS) to measure the effectiveness of our SMC method. ESS of particles at time $t$ is given by $\text{ESS}_t = (\sum_n (\tilde{w}_t^{(n)})^2)^{-1}$. If $q(\boldsymbol{z}_{1:T}|\boldsymbol{x}_{1:T}) = p(\boldsymbol{z}_{1:T}|\boldsymbol{x}_{1:T})$, expected ESS is maximized and equals the number of particles (equivalently, the normalized importance weights are uniform). Note that ESS alone is not a sufficient metric, since it does not measure the absolute quality of samples, but rather the relative quality.

## 5.1 Inference in a Benchmark Nonlinear State-Space Model

In order to evaluate the effectiveness of our adaptive SMC method, we tested our method on a standard nonlinear state-space model often used to benchmark SMC algorithms [2, 3]. The model is given by Eq. 3, where $\theta = (\sigma_v, \sigma_w)$. The posterior distribution $p_\theta(\boldsymbol{z}_{1:T}|\boldsymbol{x}_{1:T})$ is highly multi-modal due to uncertainty about the signs of the latent states.

$$
\begin{aligned}
p(\boldsymbol{z}_t|\boldsymbol{z}_{t-1}) &= \mathcal{N}(\boldsymbol{z}_t; f(\boldsymbol{z}_{t-1}, t), \sigma_v^2),\ p(\boldsymbol{z}_1) = \mathcal{N}(\boldsymbol{z}_1; 0, 5), \\
p(\boldsymbol{x}_t|\boldsymbol{z}_t) &= \mathcal{N}(\boldsymbol{x}_t; g(\boldsymbol{z}_{t-1}), \sigma_w^2), \\
f(\boldsymbol{z}_{t-1}, t) &= \boldsymbol{z}_{t-1}/2 + 25\boldsymbol{z}_{t-1}/(1 + \boldsymbol{z}_{t-1}^2) + 8\cos(1.2t), \quad g(\boldsymbol{z}_t) = \boldsymbol{z}_t^2/20
\end{aligned}
\tag{3}
$$

The experiments investigated how the new proposal adaptation method performed in comparison to standard methods including the bootstrap filter, EKPF, and UKPF. In particular, we were interested in the following questions: Do rich multi-modal proposals improve inference? For this we compared a Gaussian proposal with a diagonal Gaussian to a mixture density network with three components (-MD-). Does a recurrent parameterization of the proposal help? For this we compared a non-recurrent neural network with 100 hidden units (-NN-) to a recurrent neural network with 50 LSTM units (-RNN-). Can injecting information about the prior dynamics into the proposal improve performance (similar in spirit to [15] for variational methods)? To assess this, we parameterized proposals for $v_t$ (process noise) instead of $z_t$ (-f-), and let the proposal have access to the prior dynamics $f(z_{t-1}, t)$ .

For all experiments, the parameters in the non-linear state-space model were fixed to $(\sigma_v, \sigma_w) = (\sqrt{10}, 1)$. Adaptation of the proposal was performed on 1000 samples from the generative process at each iteration. Results are summarized in Fig. 1 and Table 1 (see supplementary material for additional results). Average run times for the algorithms over a sequence of length 1000 were: 0.782s bootstrap, 12.1s EKPF, 41.4s UPF, 1.70s NN-NASMC, and 2.67s RNN-NASMC, where EKPF and UPF implementations are provided by [5]. Although these numbers should only be taken as a guide as the implementations had differing levels of acceleration.

The new adaptive proposal methods significantly outperform the bootstrap, EKPF, and UPF methods, in terms of ESS, RMSE and the variance in the LML estimates. The multi-modal proposal outperforms a simple Gaussian proposal (compare RNN-MD-f to RNN-f) indicating multi-modal proposals can improve performance. Moreover, the RNN outperforms the non-recurrent NN (compare RNN to NN). Although the proposal models can effectively learn the transition function, injecting information about the prior dynamics into the proposal does help (compare RNN-f to RNN). Interestingly, there is no clear cut winner between the EKPF and UPF, although the UPF does return LML estimates that have lower variance [5]. All methods converged to similar LMLs that were close to the values computed using large numbers of particles indicating the implementations are correct.

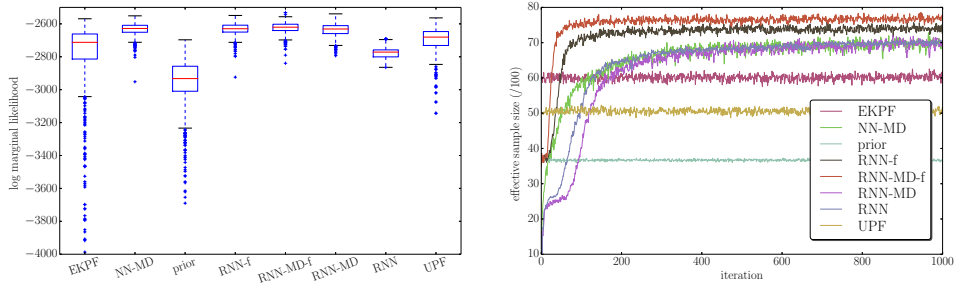

Figure 1: **Left**: Box plots for LML estimates from iteration 200 to 1000. **Right**: Average ESS over the first 1000 iterations.

|          | ESS (iter) |      | LML   |      | RMSE  |       |
|----------|------------|------|-------|------|-------|-------|
|          | mean       | std  | mean  | std  | mean  | std   |
| prior    | 36.66      | 0.25 | -2957 | 148  | 3.266 | 0.578 |
| EKPF     | 60.15      | 0.83 | -2829 | 407  | 3.578 | 0.694 |
| UPF      | 50.58      | 0.63 | -2696 | 79   | 2.956 | 0.629 |
| RNN      | 69.64      | 0.60 | -2774 | 34   | 3.505 | 0.977 |
| RNN-f    | 73.88      | 0.71 | -2633 | 36   | 2.568 | 0.430 |
| RNN-MD   | 69.25      | 1.04 | -2636 | 40   | 2.612 | 0.472 |
| RNN-MD-f | **76.71**  | 0.68 | -2622 | 32   | **2.509** | 0.409 |
| NN-MD    | 69.39      | 1.08 | -2634 | 36   | 2.731 | 0.608 |

Table 1: **Left, Middle**: Average ESS and log marginal likelihood estimates over the last 400 iterations. **Right**: The RMSE over 100 new sequences with no further adaptation.

## 5.2 Inference in the Cart and Pole System

As a second and more physically meaningful system we considered a cart-pole system that consists of an inverted pendulum that rests on a movable base [16]. The system was driven by a white noise input. An ODE solver was used to simulate the system from its equations of motion. We considered the problem of inferring the true position of the cart and orientation of the pendulum (along with their derivatives and the input noise) from noisy measurements of the location of the tip of the pole. The results are presented in Fig. 2. The system is significantly more intricate than the model in Sec. 5.1, and does not directly admit the usage of EKPF or UPF. Our RNN-MD proposal model successfully learns good proposals without any direct access to the prior dynamics.

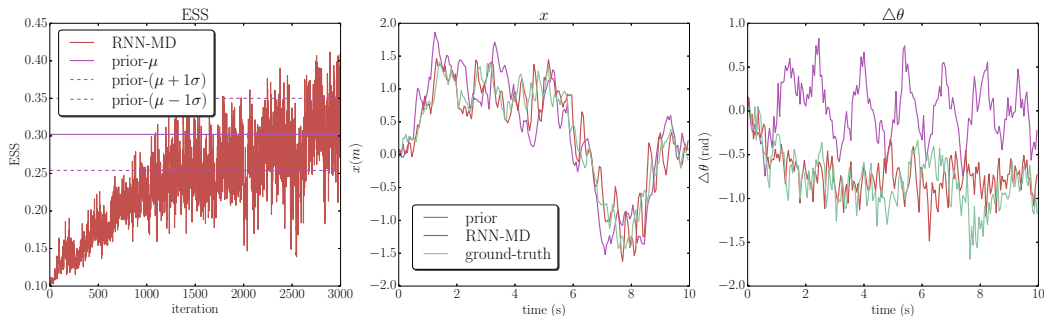

Figure 2: **Left**: Normalized ESS over iterations. **Middle, Right**: Posterior mean vs. ground-truth for $x$, the horizontal location of the cart, and $\triangle\theta$, the change in relative angle of the pole. RNN-MD learns to have higher ESS than the prior and more accurately estimates the latent states.

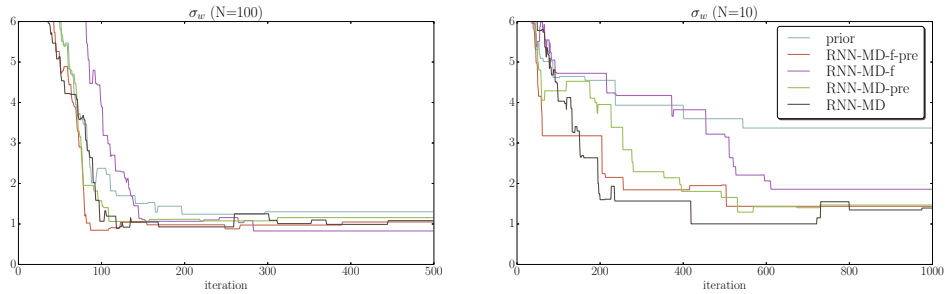

Figure 3: PMMH samples of $\sigma_w$ values for $N = \{100, 10\}$ particles. For small numbers of particles (right) PMMH is very slow to burn in and mix when proposing from the prior distribution due to the large variance in the marginal likelihood estimates it returns.

## 5.3 Bayesian learning in a Nonlinear SSM

SMC is often employed as an inner loop of a more complex algorithm. One prominent example is Particle Markov Chain Monte Carlo [3], a class of methods that sample from the joint posterior over model parameters $\theta$ and latent state trajectories, $p(\theta, \boldsymbol{z}_{1:T}|\boldsymbol{x}_{1:T})$. Here we consider the Particle Marginal Metropolis-Hasting sampler (PMMH). In this context SMC is used to construct a proposal distribution for a Metropolis-Hasting (MH) accept/reject step. The proposal is formed by sampling a proposed set of parameters e.g. by perturbing the current parameters using a Gaussian random walk, then SMC is used to sample a proposed set of latent state variables, resulting in a joint proposal $q(\theta^*, \boldsymbol{z}_{1:T}^*|\theta, \boldsymbol{z}_{1:T}) = q(\theta^*|\theta)p_{\theta^*}(\boldsymbol{z}_{1:T}^*|\boldsymbol{x}_{1:T})$. The MH step uses the SMC marginal likelihood estimates to determine acceptance. Full details are given in the supplementary material.

In this experiment, we evaluate our method in a PMMH sampler on the same model from Section 5.1 following [3].[2] A random walk proposal is used to sample $\theta = (\sigma_v, \sigma_w)$, $q(\theta^*|\theta) = \mathcal{N}(\theta^*|\theta, \mathrm{diag}([0.15, 0.08]))$. The prior over $\theta$ is set as $\mathcal{IG}(0.01, 0.01)$. $\theta$ is initialized as $(10, 10)$, and the PMMH is run for 500 iterations.

Two of the adaptive models considered section 5.1 are used for comparison (RNN-MD and RNN-MD-f) , where "-pre-" models are pre-trained for 500 iterations using samples from the initial $\theta = (10, 10)$. The results are shown in Fig. 3 and were typical for a range of parameter settings. Given a sufficient number of particles ($N = 100$), there is almost no difference between the prior proposal and our method. However, when the number of particles gets smaller ($N = 10$), NASMC enables significantly faster burn-in to the posterior, particularly on the measurement noise $\sigma_w$ and, for similar reasons, NASMC mixes more quickly. The limitation with the NASMC-PMMH is that the model needs to continuously adapt as the global parameter is sampled, but note this is still not as costly as adapting on a particle-by-particle basis as is the case for the EKPF and UPF.

## 5.4 Polyphonic Music Generation

Finally, the new method is used to train a latent variable recurrent neural network (LV-RNN) for modelling four polymorphic music datasets of varying complexity [17]. These datasets are often used to benchmark RNN models because of their high dimensionality and the complex temporal dependencies involved at different time scales [17, 18, 19]. Each dataset contains at least 7 hours of polyphonic music with an average polyphony (number of simultaneous notes) of 3.9 out of 88. LV-RNN contains a recurrent neural network with LSTM layers that is driven by i.i.d. stochastic latent variables ($\boldsymbol{z}_t$) at each time-point and stochastic outputs ($\boldsymbol{x}_t$) that are fed back into the dynamics (full details in the supplementary material). Both the LSTM layers in the generative and proposal models are set as 1000 units and Adam [20] is used as the optimizer. The bootstrap filter is compared to the new adaptive method (NASMC). 10 particles are used in the training. The hyperparameters are tuned using the validation set [17]. A diagonal Gaussian output is used in the proposal model, with an additional hidden layer of size 200. The log likelihood on the test set, a standard metric for comparison in generative models [18, 21, 19], is approximated using SMC with 500 particles.

The results are reported in Table 2.[3] The adaptive method significantly outperforms the bootstrap filter on three of the four datasets. On the piano dataset the bootstrap method performs marginally better. In general, the NLLs for the new methods are comparable to the state-of-the-art although detailed comparison is difficult as the methods with stochastic latent states require approximate marginalization using importance sampling or SMC.

| Dataset | LV-RNN (NASMC) | LV-RNN (Bootstrap) | STORN (SGVB) | FD-RNN | sRNN | RNN-NADE |
|---|---|---|---|---|---|---|
| Piano-midi-de | 7.61 | 7.50 | **7.13** | 7.39 | 7.58 | 7.03 |
| Nottingham | **2.72** | 3.33 | 2.85 | 3.09 | 3.43 | 2.31 |
| MuseData | 6.89 | 7.21 | **6.16** | 6.75 | 6.99 | 5.60 |
| JSBChorales | **3.99** | 4.26 | 6.91 | 8.01 | 8.58 | 5.19 |

Table 2: Estimated negative log likelihood on test data. "FD-RNN" and "STORN" are from [19], and "sRNN" and "RNN-NADE" are results from [18].

## 6 Comparison of Variational Inference to the NASMC approach

There are several similarities between NASMC and Variational Free-energy methods that employ recognition models. Variational Free-energy methods refine an approximation $q_\phi(z|x)$ to the posterior distribution $p_\theta(z|x)$ by optimising the exclusive (or variational) KL-divergence $\text{KL}[q_\phi(z|x)||p_\theta(z|x)]$. It is common to approximate this integral using samples from the approximate posterior [21, 22, 23]. This general approach is similar in spirit to the way that the proposal is adapted in NASMC, except that the inclusive KL-divergence is employed $\text{KL}[p_\theta(z|x)||q_\phi(z|x)]$ and this entails that sample based approximation requires simulation from the true posterior. Critically, NASMC uses the approximate posterior as a proposal distribution to construct a more accurate posterior approximation. The SMC algorithm therefore can be seen as correcting for the deficiencies in the proposal approximation. We believe that this can lead to significant advantages over variational free-energy methods, especially in the time-series setting where variational methods are known to have severe biases [24]. Moreover, using the inclusive KL avoids having to compute the entropy of the approximating distribution which can prove problematic when using complex approximating distributions (e.g. mixtures and heavy tailed distributions) in the variational framework. There is a close connection between NASMC and the wake-sleep algorithm [25] . The wake-sleep algorithm also employs the inclusive KL divergence to refine a posterior approximation and recent generalizations have shown how to incorporate this idea into importance sampling [26]. In this context, the NASMC algorithm extends this work to SMC.

## 7 Conclusion

This paper developed a powerful method for adapting proposal distributions within general SMC algorithms. The method parameterises a proposal distribution using a recurrent neural network to model long-range contextual information, allows flexible distributional forms including mixture density networks, and enables efficient training by stochastic gradient descent. The method was found to outperform existing adaptive proposal mechanisms including the EKPF and UPF on a standard SMC benchmark, it improves burn in and mixing of the PMMH sampler, and allows effective training of latent variable recurrent neural networks using SMC. We hope that the connection between SMC and neural network technologies will inspire further research into adaptive SMC methods. In particular, application of the methods developed in this paper to adaptive particle smoothing, high-dimensional latent models and adaptive PMCMC for probabilistic programming are particular exciting avenues.

**Acknowledgments**

SG is generously supported by Cambridge-Tübingen Fellowship, the ALTA Institute, and Jesus College, Cambridge. RET thanks the EPSRC (grants EP/G050821/1 and EP/L000776/1). We thank Theano developers for their toolkit, the authors of [5] for releasing the source code, and Roger Frigola, Sumeet Singh, Fredrik Lindsten, and Thomas Schön for helpful suggestions on experiments.

## Footnotes

[1]More advanced implementations resample only when the effective sample size falls below a threshold [2].

[2]Only the prior proposal is compared, since Sec. 5.1 shows the advantage of our method over EKPF/UPF.

[3]Results for RNN-NADE are separately provided for reference, since this is a different model class.

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
