[Supplementary Material · nasmc-supplementary.pdf]

# Neural Adaptive Sequential Monte Carlo
# Supplementary Material

**Shixiang Gu**[†‡]        **Zoubin Ghahramani**[†]        **Richard E. Turner**[†]
[†] University of Cambridge, Department of Engineering, Cambridge UK
[‡] MPI for Intelligent Systems, Tübingen, Germany
sg717@cam.ac.uk, zoubin@eng.cam.ac.uk, ret26@cam.ac.uk

## 1   Sequential Monte Carlo

This section reviews the basic SMC algorithm, beginning by recapitulating the setup described in the main text. Consider a probabilistic model comprising (possibly multi-dimensional) hidden and observed states $\boldsymbol{z}_{1:T}$ and $\boldsymbol{x}_{1:T}$ respectively, whose joint distribution factorizes as $p(\boldsymbol{z}_{1:T}, \boldsymbol{x}_{1:T}) = p(\boldsymbol{z}_1)p(\boldsymbol{x}_1|\boldsymbol{z}_1)\prod_{t=2}^{T} p(\boldsymbol{z}_t|\boldsymbol{z}_{1:t-1})p(\boldsymbol{x}_t|\boldsymbol{z}_{1:t}, \boldsymbol{x}_{1:t-1})$. This general form subsumes common state-space models, such as Hidden Markov Models (HMMs), as well as non-Markovian models for the hidden state, such as Gaussian processes.

The goal of the sequential importance sampler is to approximate the posterior distribution over the hidden state sequence, $p(\boldsymbol{z}_{1:T}|\boldsymbol{x}_{1:T}) \approx \sum_{n=1}^{N} \tilde{w}_t^{(n)} \delta(\boldsymbol{z}_{1:T} - \boldsymbol{z}_{1:T}^{(n)})$, through a weighted set of $N$ sampled trajectories drawn from a simpler proposal distribution $\{z_{1:T}^{(n)}\}_{n=1:N} \sim q(\boldsymbol{z}_{1:T}|\boldsymbol{x}_{1:T})$.

More specifically, consider approximating expectations of a function $\phi(\boldsymbol{z}_{1:T})$ under the unnormalised posterior distribution, $p(\boldsymbol{z}_{1:T}, \boldsymbol{x}_{1:T}) = Zp(\boldsymbol{z}_{1:T}|\boldsymbol{x}_{1:T})$, using importance sampling:

$$\int \phi(\boldsymbol{z}_{1:T})p(\boldsymbol{z}_{1:T}, \boldsymbol{x}_{1:T})\mathrm{d}\boldsymbol{z}_{1:T} = \int \phi(\boldsymbol{z}_{1:T})\frac{q(\boldsymbol{z}_{1:T})}{q(\boldsymbol{z}_{1:T})}p(\boldsymbol{z}_{1:T}, \boldsymbol{x}_{1:T})\mathrm{d}\boldsymbol{z}_{1:T} \tag{1}$$

$$\approx \frac{1}{N}\sum_{n=1}^{N} \phi(\boldsymbol{z}_{1:T}^{(n)})\frac{p(\boldsymbol{z}_{1:T}^{(n)}, \boldsymbol{x}_{1:T})}{q(\boldsymbol{z}_{1:T}^{(n)})} \quad \text{where} \quad \boldsymbol{z}_{1:T}^{(n)} \sim q(\boldsymbol{z}_{1:T}^{(n)}) \tag{2}$$

The importance weights are $w(\boldsymbol{z}_{1:T}^{(n)}) = \frac{p(\boldsymbol{z}_{1:T}^{(n)}, \boldsymbol{x}_{1:T})}{q(\boldsymbol{z}_{1:T}^{(n)})}$.

Any form of proposal distribution can be used in principle, but a particularly convenient one takes the same factorisation as the true posterior $q(\boldsymbol{z}_{1:T}|\boldsymbol{x}_{1:T}) = q(\boldsymbol{z}_1|\boldsymbol{x}_1)\prod_{t=2}^{T} q(\boldsymbol{z}_t|\boldsymbol{z}_{1:t-1}, \boldsymbol{x}_{1:t})$. Substituting this and model into the expression for the importance weights gives,

$$w(\boldsymbol{z}_{1:T}^{(n)}) = \frac{p(\boldsymbol{z}_1^{(n)})p(\boldsymbol{x}_1|\boldsymbol{z}_1^{(n)})\prod_{t=2}^{T} p(\boldsymbol{z}_t^{(n)}|\boldsymbol{z}_{1:t-1}^{(n)})p(\boldsymbol{x}_t|\boldsymbol{z}_{1:t}^{(n)}, \boldsymbol{x}_{1:t-1})}{q(\boldsymbol{z}_1^{(n)}|\boldsymbol{x}_1)\prod_{t=2}^{T} q(\boldsymbol{z}_t^{(n)}|\boldsymbol{z}_{1:t-1}^{(n)}, \boldsymbol{x}_{1:t})} \tag{3}$$

$$= w(\boldsymbol{z}_{1:T-1}^{(n)})\frac{p(\boldsymbol{z}_T^{(n)}|\boldsymbol{z}_{1:T-1}^{(n)})p(\boldsymbol{x}_T|\boldsymbol{z}_{1:T}^{(n)}, \boldsymbol{x}_{1:T-1})}{q(\boldsymbol{z}_T^{(n)}|\boldsymbol{z}_{1:T-1}^{(n)}, \boldsymbol{x}_{1:T})} \tag{4}$$

The normalising constant $Z$ can be approximated by substituting $\phi(\boldsymbol{z}_{1:T}) = 1$ into the above, which yields $Z \approx \frac{1}{N}\sum_{n=1}^{N} w(\boldsymbol{z}_{1:T}^{(n)})$. Expectations with respect to the normalised density can therefore be computed using importance weights

$$w(\boldsymbol{z}_{1:T}^{(n)})/Z = w(\boldsymbol{z}_{1:T}^{(n)})/\left[\frac{1}{N}\sum_{n=1}^{N} w(\boldsymbol{z}_{1:T}^{(n)})\right] = N\tilde{w}(\boldsymbol{z}_{1:T}^{(n)}). \tag{5}$$

So that expectations wrt the posterior can be approximated using the normalised weights $\tilde{w}(\boldsymbol{z}_{1:T}^{(n)})$, $\int \phi(\boldsymbol{z}_{1:T})p(\boldsymbol{z}_{1:T}|\boldsymbol{x}_{1:T})\mathrm{d}\boldsymbol{z}_{1:T} \approx \sum_{n=1}^{N} \phi(\boldsymbol{z}_{1:T}^{(n)})\tilde{w}(\boldsymbol{z}_{1:T}^{(n)})$. This leads to the expressions in the main paper:

$$w(\boldsymbol{z}_{1:T}^{(n)}) = \frac{p(\boldsymbol{z}_{1:T}^{(n)}, \boldsymbol{x}_{1:T})}{q(\boldsymbol{z}_{1:T}^{(n)}|\boldsymbol{x}_{1:T})}, \ \ \tilde{w}(\boldsymbol{z}_{1:T}^{(n)}) = \frac{w(\boldsymbol{z}_{1:T}^{(n)})}{\sum_n w(\boldsymbol{z}_{1:T}^{(n)})} \propto \tilde{w}(\boldsymbol{z}_{1:T-1}^{(n)})\frac{p(\boldsymbol{z}_T^n|\boldsymbol{z}_{1:T-1}^{(n)})p(\boldsymbol{x}_T|\boldsymbol{z}_{1:T}^{(n)}, \boldsymbol{x}_{1:T-1})}{q(\boldsymbol{z}_T^{(n)}|\boldsymbol{z}_{1:T-1}^{(n)}, \boldsymbol{x}_{1:T})}$$

The SIS is elegant as the weights can be computed in sequential fashion using a single forward pass along with the samples. However, such a naïve application of SIS, the distribution of importance weights often become more skewed as $t$ increases, with many samples having very low weight. To alleviate the problem, a resampling step is introduced [1] that resamples $\boldsymbol{z}_t^{(n)}$ at time $t$ from multinomial distribution given by $\tilde{w}(\boldsymbol{z}_{1:t}^{(n)})$. This replaces degenerated particles with the ones with substantial importance weights, and leads to less skewed importance weight distributions.

The SMC is given by Algorithm 1. $\gamma(\cdot|w)$ represents a multinomial distribution given by $w$, $\boldsymbol{z}_{1:t}^{(i)} = \{\boldsymbol{z}_{1:t-1}^{A_{t-1}^{(i)}}, \boldsymbol{z}_t^{(i)}\}$, $\boldsymbol{z}_{1:t}^{A_t^{(i)}} = \{\boldsymbol{z}_1^{a_1^{(i)}}, ..., \boldsymbol{z}_t^{a_t^{(i)}}\}$, and $w_t^{(i)} = w(\boldsymbol{z}_{1:t}^{(i)})$.

---

**Algorithm 1** Sequential Monte Carlo

---

**Require:** proposal: $q(\boldsymbol{z}_{1:T}|\boldsymbol{x}_{1:T})$, observation: $\boldsymbol{x}_{1:T}$, number of particles: $N$
    **for** $i = 1 : N$ **do**
        $\boldsymbol{z}_1^{(i)} \sim q(\boldsymbol{z}_1|\boldsymbol{x}_1), \quad w_1^{(i)} = \frac{p(\boldsymbol{z}_1)}{q(\boldsymbol{z}_1|\boldsymbol{x}_1)}$
    **end for**
    $\tilde{w}_1^{(i)} = \frac{w_1^{(i)}}{\sum_i w_1^{(i)}}$
    **for** $t = 2 : T$ **do**
      **for** $i = 1 : N$ **do**
        $a_{t-1,t-1}^{(i)} \sim \gamma(\cdot|\tilde{w}_{t-1}^{(i)}), \quad \boldsymbol{z}_t^{(i)} \sim q(\boldsymbol{z}_t|\boldsymbol{z}_{1:t-1}^{A_{t-1}^{(i)}}, \boldsymbol{x}_{1:t}), \quad w_t^{(i)} = \frac{p(\boldsymbol{z}_t^{(i)}|z_{1:t-1}^{A_{t-1}^{(i)}})p(\boldsymbol{x}_t|\boldsymbol{z}_{1:t}^{(i)}, \boldsymbol{x}_{1:t-1})}{q(\boldsymbol{z}_t^{(i)}|\boldsymbol{z}_{1:t-1}^{A_{t-1}^{(i)}}, \boldsymbol{x}_{1:t})}$
      **end for**
      $\tilde{w}_t^{(i)} = \frac{w_t^{(i)}}{\sum_i w_t^{(i)}}$
    **end for**

---

## 2  Adapting Proposals by Descending the Inclusive KL Divergence

This section presents the gradient derivation for learning the proposal distribution. The first approximation replaces the smoothing distribution by the filtering distribution and the second replaces analytic integration by the SMC approximation. The gradients can be computed efficiently using back-propagation through particles.

$$
\begin{aligned}
-\frac{\partial}{\partial\phi}\mathrm{KL}[p_\theta(\boldsymbol{z}_{1:T}|\boldsymbol{x}_{1:T})||q_\phi(\boldsymbol{z}_{1:T}|\boldsymbol{x}_{1:T})] &= \int p_\theta(z_{1:T}|x_{1:T})\frac{\partial}{\partial\phi}\log q_\phi(z_{1:T}|x_{1:T})dz_{1:T} \\
&\approx \sum_t \int p_\theta(z_{1:t}|x_{1:t})\frac{\partial}{\partial\phi}\log q_\phi(z_t|x_{1:t}, z_{1:t-1})dz_{1:t} \\
&\approx \sum_t \sum_i \tilde{w}_t^{(i)}\frac{\partial}{\partial\phi}\log q_\phi(z_t^{(i)}|x_{1:t}, z_{1:t-1}^{A_{t-1}^{(i)}})
\end{aligned}
$$

(6)

Figure 1: Long Short-Term Memory cell [2]

# 3 Maximum likelihood learning of parameters in latent dynamical systems

This section presents the full derivation of the parameter learning equations. Consider computing the gradient of the log-likelihood of the parameters,

$$
\begin{aligned}
\frac{\partial}{\partial \theta} \log[p_\theta(\boldsymbol{x}_{1:T})] &= \frac{\partial}{\partial \theta} \log \int p_\theta(\boldsymbol{z}_{1:T}, \boldsymbol{x}_{1:T}) d\boldsymbol{z}_{1:T} = \frac{1}{p_\theta(\boldsymbol{x}_{1:T})} \int \frac{\partial}{\partial \theta} p_\theta(\boldsymbol{z}_{1:T}, \boldsymbol{x}_{1:T}) d\boldsymbol{z}_{1:T} \\
&= \frac{1}{p_\theta(\boldsymbol{x}_{1:T})} \int \frac{\partial}{\partial \theta} \exp(\log p_\theta(\boldsymbol{z}_{1:T}, \boldsymbol{x}_{1:T})) d\boldsymbol{z}_{1:T} \\
&= \frac{1}{p_\theta(\boldsymbol{x}_{1:T})} \int p_\theta(\boldsymbol{z}_{1:T}, \boldsymbol{x}_{1:T}) \frac{\partial}{\partial \theta} \log p_\theta(\boldsymbol{z}_{1:T}, \boldsymbol{x}_{1:T}) d\boldsymbol{z}_{1:T} \\
&= \int p_\theta(\boldsymbol{z}_{1:T}|\boldsymbol{x}_{1:T}) \frac{\partial}{\partial \theta} \sum_t \log p_\theta(\boldsymbol{x}_t, \boldsymbol{z}_t|\boldsymbol{x}_{1:t-1}, \boldsymbol{z}_{1:t-1}) d\boldsymbol{z}_{1:T} \\
&\approx \sum_t \int p_\theta(\boldsymbol{z}_{1:t}|\boldsymbol{x}_{1:t}) \frac{\partial}{\partial \theta} \log p_\theta(\boldsymbol{x}_t, \boldsymbol{z}_t|\boldsymbol{x}_{1:t-1}, \boldsymbol{z}_{1:t-1}) d\boldsymbol{z}_{1:t} \\
&\approx \sum_t \sum_i \tilde{w}_t^{(i)} \frac{\partial}{\partial \theta} \log p_\theta(\boldsymbol{x}_t, \boldsymbol{z}_t^{(i)}|\boldsymbol{x}_{1:t-1}, \boldsymbol{z}_{1:t-1}^{A_{t-1}^{(i)}})
\end{aligned}
$$

# 4 Flexible and Trainable Proposal Distributions Using Neural Networks

In this section, we describe details on the network architectures used in our experiments.

## 4.1 Long Short-Term Memory

Long Short-Term Memory (LSTM) is a specific parametrization of a Recurrent Neural Network (RNN) [3, 2]. As shown in Figure 1, LSTM has an internal memory cell with separate write, read, and forget controls that are context dependent. The formulation is given by Eq. 7, where $u_t, h_t, c_t$ are input, output, and cell state of a LSTM layer at time $t$, $F(\cdot), G(\cdot)$ are activation functions, $\sigma(\cdot)$ is sigmoid activation. $W_*, B_*$ are weight matrices and bias vectors. Tanh activation is used for $F, G$ in our experiments. $o_t$ in Eq. 7 is slightly different from the standard formulation in [2], but improves computational efficiency.

$$
\begin{aligned}
c_t &= i_t \cdot F(W_u \cdot u_t + W_h \cdot h_{t-1} + B) + f_t \cdot c_{t-1} \\
h_t &= o_t \cdot G(c_t) \\
i_t &= \sigma(W_{iu} \cdot u_t + W_{ih} \cdot h_{t-1} + W_{ic} \cdot c_{t-1} + B_i) \\
f_t &= \sigma(W_{fu} \cdot u_t + W_{fh} \cdot h_{t-1} + W_{fc} \cdot c_{t-1} + B_f) \\
o_t &= \sigma(W_{ou} \cdot u_t + W_{oh} \cdot h_{t-1} + W_{oc} \cdot c_{t-1} + B_o)
\end{aligned}
\tag{7}
$$

## 4.2 Latent Variable Recurrent Neural Network and Recurrent Proposal Network

A latent variable recurrent neural network (LV-RNN) is given by $p_\theta(\boldsymbol{z}_{1:T}, \boldsymbol{x}_{1:T}) = \prod_{t=1}^{T} p_\theta(\boldsymbol{z}_t|\boldsymbol{z}_{1:t-1}) p_\theta(\boldsymbol{x}_t|\boldsymbol{z}_{1:t}, \boldsymbol{x}_{1:t-1})$, where multi-layer perceptions (MLPs) are used to learn complex non-linear mappings that parametrize conditional distrbutions. We let $p_\theta(\boldsymbol{z}_t|\cdot) = p(\boldsymbol{z}_t) =$

Figure 2: **Left**: A realization of LV-RNN, $p_\theta(\boldsymbol{z}_{1:T}, \boldsymbol{x}_{1:T}) = \prod_{t=1}^{T} p_\theta(z_t) p_\theta(\boldsymbol{x}_t | \boldsymbol{z}_{1:t}, \boldsymbol{x}_{1:t-1})$. **Right**: A recurrent proposal model, $q_\phi(\boldsymbol{z}_{1:T} | \boldsymbol{x}_{1:T}) = \prod_{t=1}^{T} q_\phi(\boldsymbol{z}_t | \boldsymbol{z}_{1:t-1}, \boldsymbol{x}_{1:t})$.

$\mathcal{N}(\boldsymbol{z}_t | 0, I)$ [1] and $p_\theta(\boldsymbol{x}_t | \boldsymbol{z}_{1:t}, \boldsymbol{x}_{1:t-1})$ be appropriate parametrization for the data . A recurrent proposal model is similarly given by $q_\phi(\boldsymbol{z}_{1:T} | \boldsymbol{x}_{1:T}) = \prod_{t=1}^{T} q_\phi(\boldsymbol{z}_t | \boldsymbol{z}_{1:t-1}, \boldsymbol{x}_{1:t})$. Figure 2 shows examples of a LV-RNN and a recurrent proposal model that are used in the experiment later. The output layer in $q$ paramtrizes a Gaussian $\{\mu, \Sigma\}$, or a mixture of Gaussians $\{p_i, \mu_i, \Sigma_i\}_{i=1:M}$, where $p_i$ denotes the mixture proportion of $i$-th Gaussian. Discrete latent variables, i.e. as in recurrent Sigmoid Belief Nets (SBN), are not explored in this paper, since the focus of the paper is on the adaptive SMC method; however, unlike [6, 7, 5], our method directly admits the learning of discrete LV-RNNs and their inference networks.

## 4.3 Mixture Density Network

The true posterior $p_\theta(\boldsymbol{z}_t | \boldsymbol{z}_{1:t-1}, \boldsymbol{x}_{1:t})$ is generally non-Gaussian and multi-modal, and therefore standard Gaussian parametrization [6, 7, 8] is less than ideal. One main advantage of our method over the SGVB is that the gradient for $\phi$ does not involve taking derivatives over the entropy of $q_\phi(\boldsymbol{z}_{1:T} | \boldsymbol{x}_{1:T})$. Besides reducing the variance of gradient estimates, this also enables $q_\phi(\boldsymbol{z}_{1:T} | \boldsymbol{x}_{1:T})$ to employ more complex distribution parametrization, such as mixture or heavy-tail distributions that can better approximate non-Gaussian distributions.

Mixture density network [9] parametrizes a mixture of diagonal Gaussians. Samples and log likelihood can be efficiently computed as in Eq. 8, where $M, D$ represent number of mixtures and dimension of each Gaussian, and $\mu_t^{(m)}, \sigma_t^{(m)}$ denote $m$-th Gaussian mean and variance. Assuming diagonal Gaussians, the dimension for $h_t$ is $M + M * 2 * D$. $l_t$ is the log likelihood of a sample $v_t$.

$$
\begin{aligned}
h_t &= W \cdot u_t + B \\
\{\mu_t^{(m)}, \log(\sigma_t^{(m)})^2, \tilde{p}_t\} &= h_t^{(m)} \\
p_t &= \text{softmax}(\tilde{p}_t) \\
m_t &\sim \text{multinomial}(\cdot | p_t) \\
v_t &\sim \mathcal{N}(\cdot | \mu_t^{(m_t)}, (\sigma_t^{(m_t)})^2) \\
l_t &= \log \sum_{m=0}^{M-1} p_t^{(m)} p(v_t | \mu_t^{(m)}, (\sigma_t^{(m)})^2)
\end{aligned}
\tag{8}
$$

# 5 Full experimental details

## 5.1 Inference in non-linear state space model

For the non-recurrent neural network (-NN-), the context window of 5 is used, i.e. the network parametrizes $q_\phi(\boldsymbol{z}_t|\boldsymbol{x}_{t-4:t}, \boldsymbol{z}_{t-5:t-1})$, or $q_\phi(\boldsymbol{v}_t|\boldsymbol{x}_{t-4:t}, \boldsymbol{z}_{t-5:t-1}, f(\boldsymbol{z}_{t-1}, t))$ for (-f-) models. When using $q_\phi(\boldsymbol{v}_t|\cdot)$ (-f-), $f(\boldsymbol{z}_{t-1})$ is treated as an additional input at time $t$, just as $\boldsymbol{x}_t$, with separate edge weights. Mixture density networks (-MD-) have 3 mixture components. Edges weights are initialized from zero-mean Gaussians, and biases are initialized with 0, except that the biases for the units that parametrize the variance of the output distribution (Gaussian and Mixture Density Layer) are set to 5 to ensure the initial proposal distribution to have sufficient tails. 100 particles are used to estimate the gradients. $\boldsymbol{x}_{1:1000}$ is generated from the prior, and is regenerated after each pass of a sequence. Gradients are computed over 100 time steps, and the hidden states are carried over within each $\boldsymbol{x}_{1:1000}$ sequence. Adam [10] with $\alpha = 0.003$ and default values for other hyperparameters is used as the optimizer.

## 5.2 Particle Marginal Metropolis-Hasting sampler

This section describes adaptive version of Particle Marginal Metropolis-Hasting sampler, used in Bayesian learning in a Nonlinear SSM experiment.

In Bayesian framework, we would like to sample from the posterior of model parameters $\theta$ along with the latent state trajectories. The Particle Markov Chain Monte Carlo framework (PMCMC) of [11] uses SMC as a complex, high-dimensional proposal for Metropolis-Hastings. Crucially, the algorithm will leave the target distribution invariant (we refer the reader to [11] for further technical details). The Particle Marginal Metropolis-Hasting (PMMH) sampler jointly samples $\theta$ and $\boldsymbol{z}_{1:T}$ targeting $p(\theta, \boldsymbol{z}_{1:T})$ using MH steps. This is more advantageous than Particle Gibbs (PG) when $\theta$ and $\boldsymbol{z}_{1:T}$ are highly correlated. Given $p(\theta, \boldsymbol{z}_{1:T}, \boldsymbol{x}_{1:T}) = p(\theta)p(\boldsymbol{z}_{1:T}, \boldsymbol{x}_{1:T}|\theta) = p(\theta)p_\theta(\boldsymbol{z}_{1:T}, \boldsymbol{x}_{1:T})$, PMMH assumes the proposal factorizes as $q(\theta^*, \boldsymbol{z}_{1:T}^*|\theta, \boldsymbol{z}_{1:T}) = q(\theta^*|\theta)p_{\theta^*}(\boldsymbol{z}_{1:T}^*|\boldsymbol{x}_{1:T})$. When it is used as an inner loop of another algorithm, the robustness of SMC is even more crucial, and adaptive approaches seem equally well-suited in this context. Using SMC to sample from $p_{\theta^*}(\boldsymbol{z}_{1:T}^*|\boldsymbol{x}_{1:T})$, the adaptive PMMH is given in Algorithm 2.

---

**Algorithm 2** APMMH

---

**Require:** proposals: $q_\phi(\boldsymbol{z}_{1:T}|\boldsymbol{x}_{1:T})$, $q(\theta^*|\theta)$, initial $\theta$: $\theta(0)$
**Require:** observation: $\boldsymbol{x}_{1:T}$, number of particles: $N$
**Require:** number of iterations: $N_i$, number of adaptation steps per iteration $N_a$

$\quad \{\boldsymbol{z}_{1:t}^{(i)}, w_t^{(i)}, \tilde{w}_t^{(i)}\}_{i=1:N, t=1:T} \leftarrow \text{SMC}(\theta(0), \phi, N, \boldsymbol{x}_{1:T})$
$\quad \gamma(0) = \prod_t \frac{1}{N}\sum_i w_t^{(i)}, Z(0) = \{\boldsymbol{z}_{1:T}^{(i)}\}_{i=1:N}$
$\quad$**for** k=1:$N_i$ **do**
$\quad\quad$**for** l=1:$N_a$ **do**
$\quad\quad\quad \boldsymbol{z}_{1:T}^*, \boldsymbol{x}_{1:T}^* \sim p_{\theta(k-1)}(\boldsymbol{z}_{1:T}, \boldsymbol{x}_{1:T})$
$\quad\quad\quad \{\boldsymbol{z}_{1:t}^{(i)}, w_t^{(i)}, \tilde{w}_t^{(i)}\}_{i=1:N, t=1:T} \leftarrow \text{SMC}(\theta(k-1), \phi, N, \boldsymbol{x}_{1:T}^*)$
$\quad\quad\quad \triangle\phi = \sum_t \sum_i \tilde{w}_t^{(i)} \frac{\partial}{\partial\phi} \log q_\phi(\boldsymbol{z}_t^{(i)}|\boldsymbol{x}_{1:t}^*, \boldsymbol{z}_{1:t-1}^{A_{t-1}^{(i)}})$
$\quad\quad\quad \phi \leftarrow \text{Optimize}(\phi, \triangle\phi)$
$\quad\quad$**end for**
$\quad\quad \theta^* \sim q(\theta^*|\theta(k-1))$
$\quad\quad \{\boldsymbol{z}_{1:t}^{(i)}, w_t^{(i)}, \tilde{w}_t^{(i)}\}_{i=1:N, t=1:T} \leftarrow \text{SMC}(\theta^*, \phi, N, \boldsymbol{x}_{1:T})$
$\quad\quad \gamma^* = \prod_t \frac{1}{N}\sum_i w_t^{(i)}$
$\quad\quad$with probability $\min[1, \frac{p(\theta^*)\gamma^* q(\theta(k-1)|\theta^*)}{p(\theta^{(k-1)})\gamma(k-1)q(\theta^*|\theta(k-1))}]$,
$\quad\quad\quad$set $\theta(k) = \theta^*, \gamma(k) = \gamma^*, Z(k) = \{\boldsymbol{z}_{1:T}^{(i)}\}_{i=1:N}$,
$\quad\quad\quad$otherwise $\theta(k) = \theta(k-1), \gamma(k) = \gamma(k-1), Z(k) = Z(k-1)$
$\quad$**end for**

---

The elegance of the PMMH sampler is that it separates SMC from MH/MCMC steps. SMC is merely used to sample the latent states and evaluate the marginal likelihood given fixed model parameters. As long as the proposal distribution has sufficient support, this means that the adapting proposals do not influence the theoretical validity of the PMMH sampler. Empirically, given limited number of particles, well-adapted proposals simply lead to larger effective sample sizes, lower variance of marginal likelihood estimates, and better mixing of the global parameters. Also, given enough representational power in the proposal model, and approximate learning rate scheduling, in the limit of APMMH steps, the proposal model converges approximately to the true marginalized posterior, $\int p(\theta, \boldsymbol{z}_{1:T} | \boldsymbol{x}_{1:T}) d\theta$.

## Footnotes

[1]Note that our algorithm applies to more complex LV-RNN models, such as the ones in [4]; these are simply design choices for particular datasets explored in our experiments and making the results comparable to [5]