[Reviews · NeurIPS 2015]

Submitted by Assigned_Reviewer_1

This is a light review.

1) Can be something said about the used "filtering approximation" of the gradient? Will the proposal distribution keep improving?

Typos: Line 421: "power proposal"?
Summary: The paper provides a nice connection between learning of the proposal distribution and the wake-sleep algorithm.

Submitted by Assigned_Reviewer_2

The paper proposes to learn proposal distributions for sequential Monte Carlo (SMC) in nonlinear state-space models (SSMs). The method (NASMC) parameterizes the proposals as (recurrent) neural networks (RNNs) which are trained to fit the filtering distribution obtained with SMC. The approach is evaluated in three different setups: (a) state inference; (b)

Bayesian learning (inference) of model parameters; (c) ML parameter learning with approximate (SMC) inference as inner loop. Results suggest that the the adaptive proposals outperform naive proposals (the prior) and also techniques such as the extended Kalman Particle filter and the Unscented Particle Filter (where the latter two are applicable). For ML learning of NN SSMs the approach performs similar to recently proposed stochastic variational approaches.

The paper is well motivated and clearly written. In addressing the question of efficient, scalable inference in latent variable SSMs the paper considers an important and timely question.

The proposed solution is not extremely original given that there has recently been a large body of work on "learning inference". Even though much of this work has focused on variational approximations the specific idea of learning a NN proposal for importance sampling -- in the non-sequential case --

has been explored e.g. in ref [26], and I think the approach is a particular instance of the cross entropy method but with NNs (e.g. Rubinstein & Kroese, 2004; [9]). Nevertheless, to my knowledge the particular setup in the paper has not been explored before, and more generally, I am not aware of much of the recent "learning inference" work being applied to time series models. Also, the proposed approach seems conceptually simple and general, and therefore of practical value.

As far as I can judge the authors have put a good effort into evaluating the proposed scheme in the context of different inference problems, covering different use-cases of SMC (including relatively low-dimensional benchmark systems as well as training a stochastic RNN on music data). On the low-dimensional benchmark data the results suggest that the learned proposal distributions can indeed be advantageous, at least compared to simple proposals such as the prior. The results for learning stochastic RNNs are less clear cut, with NASMC performing roughly on par with models of the same class trained using a variational approximation to the posterior.

Two questions the evaluation leaves at least partly open is whether we can actually expect the method to scale to large state spaces and whether it provides any advantages over variational inference networks in practice. I think the experimental section could be strengthened in two ways: Firstly, by investigating how, for a known models, the quality of the learned proposal scales with the complexity of the posterior, in particular with the dimensionality of the latent space. Secondly, it I think it would be useful to include a smaller learning experiment, on which you test different proposals, and also variational inference. In the stochastic RNN learning experiments in 5.4 it seems that neither approach to train models with stochastic latent variables does especially well (compared to RNN-NADE). But the insights that we can gain from this are in my eyes somewhat limited as they confound potential issues due to the choice of model class with potential failures of inference. (A simpler learning task may also reveal the bias of VB in learning that you discuss in section 6.) In general, I think it would be useful to study and discuss limitations and failure cases of the approach in more detail.

For instance, one worry that one might have about the approach is the fact that, at least as explored in the paper, the training of the proposal relies on the samples from the posterior which is expensive to sample from exactly. Intuitively, this seems to lead to a chicken-and-egg problem: if the posterior is poorly sampled from in the first place, then a poor proposal will be learned - leading to even worse exploration of the posterior. To reduce variance (and allow online learning) the authors propose to use a samples from a filtering approximation to the true posterior as the target for the learned proposal. Again, it seems to me that this could have undesired and, importantly, difficult-to-detect consequences. Maybe these concerns are unfounded but I think it would be worth addressing them directly.

Some additional comments:

* line 192: in what sense is the approach minimizing a global cost?

* line 219: h_t is only defined in the appendix

* Experimental evaluation: when you have a multimodal posterior does it make sense to look at the difference between the posterior *mean* and the ground truth? (This question applies to both the MD parameterization as well as the unimodal proposal...)

* MD networks in section 5.1: did you check whether these networks actually learned multimodal proposals or are the mixtures used e.g. to represent heavy-tailed proposals (which could be parameterized more directly)? Furthermore, you argue that the true posterior is multimodal. It would be very instructive to see whether this multimodality is actually properly explored by NASMC.

* how exactly did you use prior knowledge about the dynamics in the parameterization of you proposal networks?

* section 5.1 the prior as proposal seems to do really poorly but using the prior dynamics in the context of a learned NN appears to give a big boost - how does that fit together? Maybe having a proposal that corresponds to the prior mean plus some heavy-tailed noise would already give you most of the gain? (I think it would be useful to include such a simple baseline.)

Also, looking at the table in the appendix the evidence in favor of RNNs over NNs seems not so clear-cut. The main boost seems to come from incorporating prior dynamics, for either network type. But the difference between RNN-MD(-f) and NN-MD(-f) is actually quite small? [I'm finding the table in the main text almost a bit misleading.] Can you provide your "ground truth" LML estimate?

* section 5.2: Why is \delta \theta inferred so poorly when the prior is used as proposal (compared to x)? Why is \theta not part of the latent state z_t (see appendix)? Maybe say why you can't apply EKPF / UPF? (Is this because they rely on gradients?) What does inference for sigma_v look like? I'd suggest including other proposal architectures in this experiment as well.

* especially in Fig. 3, but also more generally, I think that a fair comparison of NASMC and a naive proposal such as the prior needs to take into account the computational cost of learning the proposal. Using 10 particles from the prior may perform poorly, but when taking into account the cost of learning the proposal you may be able to afford relatively more samples from the prior...

* section 5.4 I assume that the generative model structure is exactly the same as for the model trained with SGVB? Otherwise, it would be hard to compare the results.

* It would be very useful to know how sensitive the approach is to the right choice of hyperparameters (learning rates, # of particles for training the proposal, relative training of proposal vs. model network, etc.). Especially in high dimensions is there a danger of the proposal overfitting early in learning?

* I think that one nice feature of SMC inference compared to SGVB (the way it is is typically implemented, i.e. with stochastic backprop) is that it can be used with discrete stochastic latent variables. Maybe it's worth pointing this out?

* line 412ff, the comment regarding entropy: you can typically use an MC approximation when the entropy is not analytically tractable

* Another reference you may want to include: Stuhlmueller et al (2013): Learning Stochastic Inverses
Summary: I think this is a pretty well written paper that proposes a relatively simple but promising approach to improving SMC-based inference in latent-variable SSMs. The demonstration of the scalability of the method could be improved.

Submitted by Assigned_Reviewer_3

This paper proposes using recurrent neural networks as proposal distributions for particle filters. The neural networks are trained to minimize the KL divergence from the true posterior to the approximation, and a simple algorithm for an unbiased approximation to the gradient of that loss is given in terms of previous samples.

This work is part of a fashionable trend of incorporating neural networks into Bayesian inference schemes as flexible, highly parameterized, trainable functions ("put a neural net on it!"). Still, the application to particle filtering is substantially novel and the ideas are well executed.

One claim that could be softened is the "efficiency of the adaptive proposal". Yes, the NASMC system doesn't require a particle-specific approximation in its inner loop, but it does require training a neural net in its outer loop! Not to mention tuning hyperparameters and all the other complexity that comes with using a neural net as a component of a larger system. Perhaps a small section on the limitations of the NASMC approach wouldn't be out of place.

Overall, the paper is very clearly written and the experiments are convincing. The algorithm itself is simple and familiar enough that is likely to find widespread use.
Summary: A solid idea, well-executed, that should find widespread application.

Submitted by Assigned_Reviewer_4

In Sequential Monte Carlo (SMC) the proposal distribution is one of the main design choices which is important no matter which flavor of the SMC that is employed. In Adaptive SMC the proposal distribution is estimated (adapted) to produce better proposals. The authors propose an algorithm for adapting the proposal when it is a neural network.

The proposal is adapted to what in the literature is referred to as the optimal proposal--conditioned that the optimal proposal lies in the family of possible proposals.

While there are no theoretical convergence results the algorithm is demonstrated on several examples with promising results.

The paper is clearly presented and easy to follow.

It is relevant to NIPS. In my view the paper brings an important extension to the (in SMC) important problem of adapting the proposal and should therefore be accepted.

Comments: * In Figure 1 on the right plot it is not clear to me how the RNN-MD-f relates to the method by Cornebise et al. A description of possible distinction would help. It would also benefit the paper if it was clearly stated that methods are distinct.

* It is helpful to see other methods (UPF etc.) they demonstrate that your approach works. It would however be interesting to see how your method compare to other adaptive methods.

NNs are flexible so they are likely to be able to compete but will they be as efficient?

* Perhaps some conclusions about convergence can be drawn from Cornebise et al for at least some of the models?

* In the literature it is common for $x$ to denote the hidden process and $z$ (or $y$) the observed process.

* In the paragraph starting at row 232 it is not easy to distinguish particles from the true hidden state.

On rows 238-239 there should not be equality but \approx?

* In the main paper you use boldface for $z$ while in the appendix you do not. is there a reason for this?

* On row 41 in the appendix I think it should say

$w^{(i)} = \frac{p(z_1^{(i)})} \{q(z_1^{(i)}\mid x_1)}$.

* Perhaps it would be more clear if the sampling was written as $z_1^{(i)\sim \{q(\cdot}\mid x_1)}$?

Alternatively,

http://www.stats.ox.ac.uk/~doucet/doucet_johansen_tutorialPF2011.pdf have notation simliar to yours; they denote the hidden states and particles by capital letters and the variables in the proposal by lowercase letters (as I am sure you know is common to do).

* Typos: 417: this this 265: The position of (-f-) is awkward

Related literature to be discussed: ---

A Stable Particle Filter in High-Dimensions BY ALEXANDROS BESKOS, DAN CRISAN, AJAY JASRA, KENGO KAMATANI, & YAN ZHOU http://arxiv.org/pdf/1412.3501.pdf - In this paper a space-time particle filter is proposed. How does that method relate to yours? ---

ADAPTIVE METHODS FOR SEQUENTIAL IMPORTANCE SAMPLING WITH APPLICATION TO STATE SPACE MODELS

JULIEN CORNEBISE, E RIC MOULINES, AND JIMMY OLSSON http://arxiv.org/pdf/0803.0054v2.pdf - The same authors has also written: Adaptive sequential Monte Carlo by means of mixture of experts

This approach is updated since the PhD thesis from 2009 (which you cite) and is worth mentioning. A comparison to your method would be interesting.
Summary: The paper proposes a novel approach to adaptive sampling in Sequential Monte Carlo. The approach is demonstrated to work. I suggest that the paper is accepted after minor changes.

Author Feedback
Author rebuttal: We would like to thank all the reviewers for their very constructive feedback.

Reviewer_1

Q1. comparisons to variational methods
A1. The main goal of the paper was to show that NASMC brings advantages to the different use cases of SMC using standard benchmarks used by this community, including non-linear state estimation (experiments 1&2) and Bayesian inference via MCMC (experiment 3). Experiment 4 was designed to show that NASMC enables SMC to be extended to larger models and datasets than the SMC community typically considers. Although not the focus of this paper, we believe that NASMC can also bring advantages over variational approaches too, but the reviewer is correct that, although the initial results here are promising, further experimentation is needed. Indeed, this question is not thoroughly resolved in the static case either [26].

Q2. does it scale to large state spaces?
A2. We agree that this is a key question and believe that NASMC has the potential to advance this frontier. The results on the MIDI (a common benchmark for this purpose [17,19]) are promising, but we agree further work is needed. Using a toy problem and varying the state dimensionality is a very sensible idea.

Q3. chicken&egg problem in learning proposals
A3. A priori we were also worried that adaptation based upon a poor proposal might lead to a poorer proposal. However, we never encountered this, possibly because we took the prudent step of initialising the proposal to have wide support. Additional steps could also be taken such as initially using a larger number of particles or initialising from the prior proposal.

Q4. does NASMC minimize a global cost?
A4. Thank you for indicating that this was not clear. The main point here is that NASMC (& [26]) differs from wake-sleep, which is known to optimize two different objectives that could interact undesirably (there is no Lyapunov function). Instead NASMC method bears resemblance to adaptive rejection sampling, in that the efficiency of the proposal improves over time, but the samples are always valid.

Q5. evaluating posterior sample quality
A5. RMSE is sensitive to multimodality. For this example, when a mode is missed the mean value changes significantly. While it's possible to design statistics that are more sensitive, we decided to use standard performance metrics from the SMC literature.

Q6. is multi-modal proposals learned in 5.2?
A6. Yes.

Q7. using prior knowledge in proposal network
A7. The proposal model (-f-) has an additional input at each time step to the neural network equal to f(z_{t-1}) (the prior dynamics).

Q8. performance of RNN vs NN
A8. NN uses context window of 5 time steps (will add details). The relatively small performance gap between RNN and NN is due to the simple Markovian nature of the benchmark nonlinear state-space model.

Q9. questions about section 5.2
A9. We will update the details on this experiment in the appendix and this shall clarify all these questions.

Q10. computational cost of learning proposal versus prior (see R2 Q1 too)
A10. After drawing the sample, the gradient computation can be parallelized over p and q. In practice we do not think learning proposal imposes slow-down in training speed if it's parallelized correctly. But it's true that proposal learning involves one more forward pass.

Reviewer_2
Q1. computational efficiency of NASMC
A1. This is a very valid point. We will add a discussion of this to the paper. As you mentioned, for adaptive PMMH, naive implementation of adaptation can become very expensive. However, the algorithm is flexible, and the RNN proposal does not need to be retrained at each step. A comparison on training scheduling (how often RNN is retrained) vs convergence/mixing of MCMC will be valuable for the particle MCMC community.

Reviewer_3

Q1. comparison to Cornebise et. al.
A1. Cornebise et. al. use the same KL objective to learn proposals of a specific form (mixture of Gaussian/Student-t) that depends on the sample at the previous time-step; NASMC admits more general parametric family that potentially depends on the entire sample trajectory. Lastly, Cornebise et al combines this form of proposal adaptation learning with another efficient sufficient-statistic based learning, which constrains the model choice. We will add a discussion of this to the paper.

Q2. compare to other adaptive methods
A2. This is an excellent addition, in particular to Cornebise et al which is the most powerful scheme we are aware of. These methods are, however, complex and require very specific implementations that was hard to unpack from the papers we knew of at submission time. Instead we chose a standard SMC benchmark to demonstrate our method against EKPF and UPF which are arguably the most widely used adaptive methods.

Q3. convergence proofs for some specific models?
A3. A great suggestion that we are working on. Building on Cornebise et. al.'s results is desirable.